# Geometric Measurements on a CNC Machining Device as an Element of Closed Door Technology

**DOI:** 10.3390/s21144852

**Published:** 2021-07-16

**Authors:** Grzegorz Bomba, Artur Ornat, Piotr Gierlak

**Affiliations:** 1Pratt & Whitney Rzeszów S.A., ul. Hetmańska 120, 35-078 Rzeszów, Poland; grzegorz.bomba@prattwhitney.com (G.B.); artur.ornat@prattwhitney.com (A.O.); 2Faculty of Mechanical Engineering and Aeronautics, Rzeszow University of Technology, al. Powstańców Warszawy 8, 35-959 Rzeszów, Poland

**Keywords:** closed door technology, CNC machining device, measurement, repeatability, reproducibility

## Abstract

The article discusses the quality testing of a measuring system consisting of a CNC machine with measuring probes. The research was conducted in a broader context regarding the implementation of the closed door technology, i.e., production without human intervention, in an aviation plant manufacturing aircraft gearbox systems. This technology may involve automated measuring operations performed in machining centers, and not in measuring laboratories, provided that the quality of the measurements is appropriate. The aim of the study was to investigate whether the CNC machining device can be used to measure the geometric features of aircraft gearbox housing. For this purpose, measurement experiments were carried out with the use of three different probes. Measurements were carried out using four sequences of increasing complexity, so that, after error analysis, it was possible to find the causes of possible irregularities. A reference ring with known dimensions and position in the working space of the machine was used for the measurements performed as part of the assessment of the measurement system. The quality of the measurements was evaluated with the use of repeatability and reproducibility testing and statistical process control. The analysis results showed that the tested measurement system ensures adequate accuracy and repeatability, and the measurement process is characterized with adequate efficiency in relation to the manufacturing tolerance of the components produced using the machine. Thus, it was proven that the measurement process can be carried out on a machining device, which enables its integration into the closed door technology.

## 1. Introduction

The concept of machining according to the closed door technology (CDT) [1] is based on making the best use of a production line built of autonomous machines, capable of continuous production with minimal human intervention. In the case of product manufacturing based on mechanical processing, such a production line generally consists of a group of computer numerically controlled (CNC) machining centers, equipped with a whole range of devices and systems ensuring process autonomy. These devices include the multi-pallet system integrating several centers into a single whole, as well as the master control system supervising the operation of the production line, pallet system, and CNC machining centers. Machine tools should be equipped with systems controlling the tool condition, e.g., by optical measurement with a laser beam or contact elements [2].

The CNC machining centers are normally equipped in a package of measuring probes operated with dedicated software. This enables carrying out measurements and saving the results in the form of CNC system variables or measurement reports.

CDT is characterized by the full use of production capabilities thanks to better planning of workshop tasks and maximum reduction of human factor impact on the production process [3,4]. With the use of closed door machining, the operators play a different role than in the case of traditional machining methods [5]. They move up the skill chain to become machining cell coordinators, capable of operating several machines simultaneously and are supposed to keep the production line running by performing process-related activities. They focus on supervision and control tasks with high added value, while the machines are operating independently.

Figure 1 shows the course of a standard machine part machining process compared to the process carried out using CDT. At the moment, the most difficult stage of the process, in view of implementing CDT, usually involves the measurement operations, i.e., the last stage of the process. In principle, they should only take place in the machining device [6,7,8]. This requires replacing the coordinate measuring machine (CMM) with a machine tool equipped with measuring probes.

The implementation of CDT is possible provided that the machine tools can be used to make reliable measurements of parts between successive machining operations and after their completion [9,10,11].

In this work, the repeatability and reproducibility test (R&R) [12] and the statistical process control (SPC) method [13] were used to assess the quality of measurements carried out in CNC machines. R&R is a statistical method used to search for the sources of the measurement process variability. The variability may depend on many factors, such as: people performing the measurement, measurement tools, measurement methods, environmental conditions, and measured objects. This method considers the analysis of the differentiation of measurements of measuring instruments (repeatability) and the fluctuation of measurements made by operators (reproducibility) [12].

SPC is a set of techniques and statistical methods used to evaluate the process stability. The purpose of SPC is to prevent nonconformities by detecting and signaling process faults. SPC responds to the ineffectiveness of traditional quality control. Instead of controlling the finished product, the quality inspectors or machine operators themselves inspect individual components. They do not wait for defective items to appear. If subsequently produced items come dangerously close to the acceptable limits (tolerance), they take necessary actions. These activities may include: notifying the superiors, adjusting the machine settings, replacing a worn machine part, etc.

SPC is focused on continuous improvement. It mostly involves preventive measures. The research on process does not only provide information regarding its course and possible deviations, but also helps to understand the causes of process variability. Thanks to systematic monitoring, the organization can minimize losses by removing the identified faults and errors on an ongoing basis. At the same time, based on information about the emerging issues, the management designs processes in a way that prevents their occurrence, e.g., using Poka Yoke [14], quality design or Failure Mode and Effects Analysis [15].

SPC is also a strategy for identifying, estimating and reducing variability in processes, products, and services. The widespread use of SPC methods and tools helps to create products and services that will constantly meet customer requirements. Most often, SPC is used in the (large series) production industry, where there is a high homogeneity of individual product features. There are also varieties of SPC methods that can be successfully used to supervise a small series or even single piece processes.

Typically, R&R or SPC methods are used to assess the quality of products or the quality of the production process. This work presents a slightly different approach. The measurement process is evaluated using a CNC machine equipped with measuring probes, with only one element being measured, that is the reference ring. Thus, the values to be obtained from the measurements are known, and it is the measurement system that is assessed instead of the product manufactured in the machine. The positive result of the measurement system assessment, in terms of the adopted criteria, allows it to be used as an element of CDT to replace a CMM used in a standard process.

Several methods are well known for evaluating machine quality, but they do not include machines with probes as measuring devices. For example, the popular ballbar test [16] is only used to evaluate the accuracy of executing programmed machine movements in a certain area of the machine’s workspace. Another way to evaluate the accuracy of a machine is to make a test part [17], whose geometric parameters are then determined, for example, on a CMM. This test determines the machine’s ability to perform its intended machining operations. In addition, it considers many factors of a dynamic nature that are not present during the measurement process. Finally, there are methods for machine calibration using laser interferometry [18]. They are used to compensate for the positioning of linear axes, which is simply motion correction. However, none of the methods mentioned are used to evaluate the machine, which is treated as a measurement system.

Evaluation of the measurement process on the machine is required to estimate measurement quality and measurement errors. Currently, norms such as ISO 230-2:2014 and ISO 230-6:2002 only consider evaluating machine repeatability on a single axis or diagonal. Therefore, attempts to find other methodologies for assessing the quality of measurements are described in the scientific literature. The efforts of some researchers are directed towards the search for universal methods, while others try to solve the problem for selected cases, and these are usually applied works. In article [9], the authors searched for a general model to evaluate measurement repeatability for a five-axis machine in any measurement situation. In articles [1,4], the authors implemented on-machine measurements to gain information about machining quality and adjust the milling process. In article [3], the authors implemented measurements on a machine to calibrate the cutting process and compensate for the effect of susceptibility. In article [6], the authors used on-machine measurements in a multi-step milling process to compensate for the susceptibility effect of thin-walled workpieces. Their results show that this approach significantly improves process quality. However, in none of these articles, and in many similar ones, the authors did not deal with the evaluation of measurement quality, but only with the use of the measurement system itself on machine. Indeed, the purpose of these approaches was the need to update workpiece data during the process, while the final control of part dimensions was not analyzed. Therefore, an analysis of the quality of the measurement systems is not presented there. The approaches discussed assume a final check on another measuring device.

For these reasons, the authors of this article have addressed the issue of evaluating the quality of a measurement system consisting of a CNC machine tool and a contact probe system. The accuracy of the machine itself or the probes themselves were not analyzed separately, as is common practice, but the system as a whole was treated. The aim of the work was to replace the CMM in the final inspection of parts by a CNC machine, which is a novelty compared to solutions in which measurements on CNC machines only serve to correct processes, without a guarantee of sufficiently high accuracy. The presented solution is not generalized, but refers to the quality control of aircraft gearbox cases and in assessing the quality of the system, reference was made to the requirements for these cases. Nevertheless, the requirements for aerospace parts are high enough that positive test results will allow them to be used in other areas of manufacturing as well.

The subsequent sections of this work describe the evaluation of the measurement system in the context of its application in the production of aircraft accessory gearboxes (AGB). Section 2 describes the process of producing the housing, Section 3 presents automatic measurements carried out on a CNC machine with the use of a reference element, as well as the measurement results. Section 4 presents the evaluation of the measurement process using the R&R method and Section 5 describes the evaluation of the measurement process using the SPC method. Section 6 includes the discussion of the results, Section 7 includes comparative study and the conclusions are set out in Section 8.

## 2. Manufacturing of Aircraft Gearbox Housing

The AGB housing (Figure 2) is intended for arranging the interrelated components, assemblies, and mechanisms. In order to reduce the weight, the external shape of the housing is closely adjusted to the outlines of the internal components. This results in the complexity of the shape and specific technological requirements. In order to minimize the weight, the housing is made of a thin-walled light alloy C355.0-T6. The type and condition of material used for the case is detailed in specification AMS4215 [19].

One of the most important geometric parameters of the gearboxes is the arrangement of the bearing seats. It has a direct impact on gear interaction, which is the main factor determining gearbox durability [20]. Machining of gear bearing seats is performed using precise machining centers. The position of the bearing seats is largely determined by the kinematic accuracy of the machine tool [17].

Additional difficulties encountered when machining such thin-walled gearbox cases are deformations resulting from changes in the stress state during the various stages of manufacturing. Aerospace gearbox cases are made as castings with process allowances. During the machining process, there are changes in the stress state resulting from: the release of stresses present in the raw material from the casting process [21] and the introduction of deformations resulting from the machining process [22]. However, this is not the subject of the research presented in this article, but more information on this topic is presented in the article [20].

To a certain extent, CNC precision machining centers and CMM, coordinate measuring machines, are characterized with similar properties. They include, among others:Kinematics, because they are mainly machines using a Cartesian coordinate system;Precise measurement systems enabling the determination of the position of a certain point on a Cartesian plane;Measuring equipment, i.e., measuring probes;Possibility of programming measuring cycles; andPossibility to generate reports directly or using system variables.

The next chapter presents the tests that allow to assess the quality of measurements performed in a CNC machining center, which can provide the answer to the question whether it is possible to carry out selected measurement operations in a machining station instead of a measurement laboratory.

## 3. Automatic Measurements in a CNC Machining Centre

In order to check the quality of the measurements performed in a machining device, a number of tests were carried out, considering various measuring probes and numerous variants of approaches with the spindle-mounted measuring probe.

The measured object was the Mitutoyo reference ring with an internal diameter of ϕ = 86,999 (mm) (Figure 3). The data regarding the reference ring axis and diameter as well as its face surface in relation to the machine reference system were entered into the measurement program.

Before starting the measurement experiments, a probe calibration procedure was performed. There are several methods for calibrating measurement probes [18,23]. One of the basic method uses a master ring gauge and a test rod to determine offsets in the X, Y, and Z axes. First set the master ring gauge relative to the machine coordinate system (Figure 4) and then take calibration measurements with each probe. There are other calibration methods that have their own advantages and can be used to eliminate some steps in the calibration procedure discussed earlier. Instead of using a test rod, it is possible to mill a surface and treat it as a Z-axis reference for the probe length calibration. However, this method is only suitable for use on machines without rotary axes. Instead of using a dial indicator to determine the center of the master ring gauge, with its inherent human and measurement errors, a hole is drilled in the part placed on the machine. The center of this hole is therefore known very precisely and can then be used by the calibration procedure to calculate the X, Y offset of the stylus ball. An additional disadvantage of methods using machining is the influence of the dynamic state of the machine and the state of the tool on the procedure. Another way to eliminate the master ring gauge is to use a calibration ball. This is especially useful when recalibration is done frequently and the ball can be screwed into a dedicated location on the fixture or machine table.

Due to the calibration standards accepted in the plant and the recommendations of the probe manufacturer, a calibration method using a test rod and a calibration master ring gauge was selected. An additional justification is that the measurement of the position of the master ring gauge diameter is the same measurement that is made when determining the position of the bearing seat axes in the gearbox cases.

First, the position of the reference ring axis in the machining tool coordinates was established (Figure 4). X_0_ and Y_0_ for the workpiece coordinate system were assumed and determined at the location of the established ring axis. Similarly, the coordinate of the position of the ring base surface was assumed as Z_0_. These coordinates were saved in the machine parameter table and the measurement errors were to be determined relative to them.

Next, each touch probe was calibrated according to Renishaw (Renishaw plc, Wotton-under-Edge, UK) probe calibration procedure [23]. For each of the calibrated probes, the probe length corrections and the so-called radial correction factors used in diameter measurement were measured and entered in the CNC machine registers.

Three RMP600 (Renishaw plc, Wotton-under-Edge, UK) strain gauge probes (Figure 5) with technical parameters characterized in Table 1 and Table 2 were tested. The advantage of such a set of probes with styluses of various lengths and tip shapes, is the ability to measure the parameters of holes with different depths and enlarged diameters at the bottom.

Objective: Measurements of the reference ring characteristics, i.e., TP position of the ring axis, ring diameter ϕ, and location of the base surface Z must reflect the measurements of analogous critical characteristics of the aircraft gearbox housing, i.e., The TP position of the axis of the housing bearing seat, the diameter of the housing bearing seat, the location of the base surface (or other surface parallel to the XY plane of the machine tool) of the aircraft gearbox housing [17].

### 3.1. Measurement Sequences

In order to assess the impact of individual elements of the measurement process on the quality of measurements, tests were carried out according to the four measurement sequences described below. The study began with the simplest sequence involving movement of only one axis, through a sequence containing movement in three axes, a sequence with movement to the end position, and finally a sequence containing an operation to retrieve the probe from the tool magazine. Thus, each next analyzed measurement sequence contained new component movements or operations. Analysis of measurement errors for successive sequences allowed us to determine the effect of individual actions and operations on measurement accuracy. The selected sequences occur in the actual manufacturing process when measuring geometric characteristics of parts, and the type of sequence depends on, among other things, the type of geometric characteristics being performed. In the following subsection, four sequences are described.

#### 3.1.1. Sequence 1

Sequence 1 involved the measurement of the base surface coordinate Z. In sequence 1 (Figure 6), the Z coordinate of the base surface was measured and defined as Z_0_ for the position of the workpiece coordinate system. At the beginning of this sequence, the measuring probe approached the Z_0_ sequence to a distance of approximately 10 mm with alignment speed. Next, the measurement of the surface coordinates was performed. In the course of the measurement, the probe moved only in the direction of the Z axis (move 1) until it touched the measured surface, and then returned to its starting position (move 2). This cycle (move 1–move 2) was repeated 25 times, resulting in 25 measurements of the Z coordinate of the base surface Z_0_ [24]. The value of the measured Z coordinate was calculated by the O9811 macro for Renishaw Okuma OSP P300M (Okuma Corporation, Ōguchi, Japan) control system.

#### 3.1.2. Sequence 2

Sequence 2 involved the measurement of the Z coordinate, as in Sequence 1, as well as the measurement of the position of the ring axis TP and of the ring diameter ϕ. In the first phase, sequence 2 included elements identical to sequence 1. First, the Z_0_ surface was approached and the 25 values of Z_0_ surface coordinates were measured, exactly as in sequence 1. After the Z_0_ measurement cycle was completed, the probe moved to point 0 whose coordinates were defined as the center of the coordinate system for the reference ring measurement. Next, an additional cycle of measurements of the diameter ϕ and the TP position of the measured diameter axis was carried out (Figure 7). After reaching the 0 point coordinates, the probe moved along the Z axis several millimeters down the measured ring.

Next, the measurements of the ring inner surface point coordinates were performed in the Y axis direction (moves 1, 2, and 3). After the probe returned to point 0, the measurements of the ring inner surface point coordinates were performed in the X axis direction (moves 4, 5, and 6). After the measurement was completed, the probe returned to its starting coordinates. The measuring cycle was repeated 25 times. Every time, the value of the measured diameter ϕ and the position of the diameter axis TP were calculated according to the Renishaw algorithm included in macro O9814 for the Okuma OSP P300M control system.

#### 3.1.3. Sequence 3

Sequence 3 involved the measurement of the Z coordinate, the position of the ring axis TP and the ring diameter ϕ including an approach of the probe from the machining tool replacement point. Sequence 3 was analogous to sequence 2 with one difference. Each measuring cycle started with an approach from the coordinates of the machining tool replacement point (moves 1, 2, 3, and 4—Figure 8a). After completing the measuring cycle as in Sequence 2, the probe returned to the machining tool replacement point (moves 5, 6, 7, and 8—Figure 8b).

#### 3.1.4. Sequence 4

Sequence 4 involved the measurement of the Z coordinate, the position of the ring axis TP and the ring diameter ϕ including probe collection from the tool warehouse before each measurement and an approach of the probe from the machining tool replacement point. Sequence 4 was analogous to sequence 3 with the following differences. Before every measurement:The tested measuring probe was returned to the tool warehouse;A machining tool was collected in order to mount a different tool holder in the machine tool spindle;The machining tool was returned to the machine tool warehouse;The appropriate tool probe was collected.

The four described sequences were defined to simulate, as closely as possible, the cases of measurement with probes used during the inspection of workpieces in the machine during the production process. The simplest case was evaluated first, with the probe located in the spindle and only one of the characteristics being measured (Sequence 1). Next, several characteristics located close to one another were measured, e.g., within one of the bearing seats, i.e., the position of the thrust bearing face, the bearing seat diameter ϕ and the position of the seat axis (Sequence 2). The next sequence (Sequence 3) represents the case in which the measuring probe was located in the spindle and several characteristics were measured, but in order to perform the measurement, it was necessary to move the spindle to a considerable distance within the machining space of the CNC machine tool. This situation occurs, for example, in the case of measuring diameters, depth and location of mounting pin holes. Due to their role, they are spaced as widely as possible, usually at the two ends of the gearbox housing. The tolerances of the position and execution of their diameters are among the narrowest in the entire housing.

Sequence 4 describes a case analogous to the one described above, however, it was extended with a cycle of returning the tested probe to the warehouse and collecting a different holder in place of the tested probe, in this case, any machining tool. After mounting the tool holder in the spindle, it was returned to the warehouse and the tested probe, mounted in its own holder, was taken from the tool warehouse and placed in the machine tool spindle. This test is intended to illustrate additional errors resulting from mounting the tested probes (probe holders) to the machine tool spindle and the impact of their repeated mounting in the spindle.

### 3.2. Measurement Results

Each measurement sequence was performed 25 times to obtain data for a statistical evaluation of the measurement process. The data is presented as graphs in Figure 9, Figure 10 and Figure 11. First, the results of measuring the reference ring base surface Z coordinate position with probes T2, T3, and T4 (Figure 9) are presented for four sequences. Next, the errors of measuring the ring axis TP position (Figure 10) and the U reference ring diameter (Figure 11) are presented. It should be kept in mind that the ring axis coordinates are X_0_ and Y_0_. These coordinates are used as reference for the ring axis measurement, so the ring axis position measurement error is equivalent to the measured value of the ring axis TP position. On the other hand, the error of measuring the diameter of the reference ring in a single measurement is determined by the following formula
(1)$$U_{i}=\left|\phi_{M}-\phi_{i}\right|$$
where ϕ_M_ = 86.999 (mm) is the reference ring diameter, and ϕ_i_—is the measured diameter value in the measurement i.

Table 3 includes maximum values of: ring base surface Z coordinate position measurement errors, ring axis TP position measurement errors and ring U diameter measurement errors.

## 4. R&R Measurement Process Evaluation

The main element of the R&R method is the determination of the R&R (repeatability and reproducibility) index, which is performed by measuring the selected product with the use of a specific measurement system. The primary purpose of performing an R&R analysis is to identify the causes of variability in the measurement process so that their impact can be interpreted and minimized. First, for the purposes of the analysis, the following mean values were determined for various sequences and probes: errors in measuring the ring base surface Z coordinate position, errors in measuring the ring axis TP position and errors in measuring the ring U diameter (Table 4).

First, the repeatability of the EV measurement system was determined. Repeatability is a change in measurements taken by one person or instrument in the same position and under the same conditions. The measurement is considered reproducible when the change is smaller than the agreed range. Repeatable measurement conditions include: a specific, standard measurement procedure, observer, the same measuring instrument or tool, used in the same conditions and place, as well as repeating the measurement in a short intervals.

The repeatability of the measurement method can be expressed quantitatively using the characteristics of results dispersion (e.g., as standard or expanded uncertainty—σ_g_). The repeatability given as expanded uncertainty is abbreviated as EV (Equipment Variation) and it is determined by means of the following correlation [25]:EV = R_AVE_K_1_(2)
where K_1_ is a constant depending on the number of operators (measuring probes)—in this case, for three measuring probes, K_1_ = 0.5908, R_AVE_ is the average value of the measurement error for three probes (Table 5, line 3). The three probes, T2, T3, and T4, are treated as three operators. Each of the operators measures three different characteristics of the reference ring, i.e., Z, TP, and U. The calculation requires the knowledge of the average values (R_AVE_) for the individual measured characteristics. They were calculated on the basis of the results presented in Table 4 as an average of the mean values for the individual measuring probes (average of the individual columns in Table 4). The EV values for the individual sequences are given in Table 5, line 4.

The next step in the analysis is to calculate the reproducibility. Reproducibility is one of the R&R method components and refers to the ability of a test or experiment to be duplicated or reproduced by other operators. The AV (Appraiser Variation) reproducibility can be quantified using the characteristics of results dispersion by means of the following correlation [25]:(3)$$\mathrm{AV}=\sqrt{{\left(X_{\mathrm{DIFF}}K_{2}\right)}^{2}-\frac{{\mathrm{EV}}^{2}}{\mathrm{nr}}}$$
where X_DIFF_ is the spread of the means for individual probes (Table 5, line 5) determined on the basis of Table 4, K_2_ is a constant depending on the number of measuring probes—in this case, for three measuring probes, K_2_ = 0.5321, n is the number of measured parts—in this case n = 1 since only 1 reference ring was measured, r is the number of tests in each sequence—in this case r = 25. The AV values for the individual sequences are given in Table 5, line 6.

Based on the EV and AV values, the R&R expanded uncertainty values were calculated by means of the correlation [25]:(4)$$R\&R=\sqrt{{\mathrm{EV}}^{2}+{\mathrm{AV}}^{2}}$$
The numerical values of the R&R index are presented in line 7 of Table 5

From the point of view of result interpretation, it is more preferable to determine the relative %EV, %AV, and %R&R indices relative to the manufacturing tolerance zones of the components to be measured. The relative indices are defined in the following manner [25]:(5)$$\%\mathrm{EV}=\frac{\mathrm{EV}}{T}100\%$$
(6)$$\%\mathrm{AV}=\frac{\mathrm{AV}}{T}100\%$$
(7)$$\%R\&R=\frac{R\&R}{T}100\%$$
where T is the scope of component tolerance zone. The manufacturing tolerance zones of particular components resulting from design requirements are presented in Table 6.

It is assumed [25] that if the analysis results are lower than 10%, they are acceptable. If the results are between 10% and 30% they are conditionally acceptable, and if the results exceed 30% they are not acceptable. If the calculated %EV and %AV values are greater than 30%, it may indicate:Incorrect handling of the instrument (%AV), i.e., improperly calibrated measuring probe;Incorrect instrument function (%EV), i.e., a poorly operating measuring system consisting of the machine tool and the measuring probe;An incorrectly selected instrument (%AV), i.e., an incorrectly selected measuring probe, in such case the accuracy of the machine tool-probe measuring system is too low;Problems with reading the instrument indications (%AV), i.e., problems with correct reading of coordinates;Improper mounting of the instrument or part (%EV), i.e., improper mounting of the probe in the spindle.

In the analyzed task, %EV values are lower than 3%, and %AV values are lower than 3.8% of the structurally required tolerance values for the tested critical characteristics.

## 5. SPC Measurement Process Evaluation

SPC activities are based on the principle of systematic collection and testing of a certain number of samples. This method is based on the requirement of distinguishing the reasons for the variability of the analyzed process. The factors affecting the stability of the process can be divided into two groups:Natural factors—they are inextricably linked with the process, they are usually numerous, but none of them is strong enough to disrupt the process (e.g., changes in atmospheric pressure, temperature, air humidity, etc.); when only these factors act on a process, it is said to be statistically controlled;Special factors—they act with great force, causing process disruption (e.g., machine failure); when they act exclusively, the process is not statistically controlled.

There are two concepts that are very strongly related to the use of SPC tools: accuracy (a measure of compliance of products with a standard) and precision (a measure of dispersion).

An important concept related to SPC methods is the process capability, i.e., the degree to which the process meets the quality requirements, assessed with the help of capability indices. If the tolerance of the tested feature is considered, the potential and actual ability of the process to meet the quality requirements can be determined. Thanks to this, it is possible to determine how many products are within the assumed specification limits.

The C_p_ and C_pk_ indices are used to assess the process capability. The C_p_ index (potential capacity)—means the precision of the process and is a measure of the width of the actual dispersion in relation to the width of the tolerance zone. It is determined by means of the following formula [26]:(8)$$C_{p}=\frac{T}{\mathrm{ND}}=\frac{\mathrm{USL}-\mathrm{LSL}}{\mathrm{UCL}-\mathrm{LCL}}=\frac{T}{6\sigma }$$
where T is the scope of the tolerance zone, ND is the process variability, USL is an upper specification limit, LSL is a lower specification limit, UCL is an upper control limit, LCL is a lower control limit, and σ is a standard deviation.

The C_pk_ index (actual capacity) is related to the accuracy of the process and considers the mutual shift of the dispersion zone relative to the tolerance zone. It is treated as a one-sided (right-sided or left-sided) indicator. In other words, the C_pk_ index evaluates the actual process capability by comparing the difference between the process center and each of the tolerance zones to a half of the natural process variability. The C_pk_ index is determined on the basis of the following formula [26]:(9)$$C_{\mathrm{pk}}=\min\left(C_{\mathrm{pL}},C_{\mathrm{pU}}\right)$$
where:(10)$$C_{\mathrm{pL}}=\frac{\bar{X}-\mathrm{LSL}}{0.5\mathrm{ND}}=\frac{\bar{X}-\mathrm{LSL}}{3\sigma }$$
(11)$$C_{\mathrm{pU}}=\frac{\mathrm{USL}-\bar{X}}{0.5\mathrm{ND}}=\frac{\mathrm{USL}-\bar{X}}{3\sigma }$$
where $\bar{X}$ is the actual value of the mean dispersion. The C_pk_ index is therefore a measure of the distance between the actual value of the mean dispersion and the center of the tolerance zone. By determining the value of the C_pk_ index, it is possible to assess both the accuracy and precision of the tested process.

The rules for diagnosing the qualitative capacity of the process based on the values of the C_p_ and C_pk_ indices are presented in Table 7 [26].

Summing up, the C_p_ and C_pk_ indices can be regarded as process capability indicators because they inform what quality the process is capable of in a short period of time. This is the highest process capability. They are compared to P_p_ and P_pk_ indices, the so-called process performance indicators, which inform about what will happen to the process beyond a short period of time.

Thus, the difference in the calculation of P_p_ and P_pk_ compared to C_p_ and C_pk_ comes down to the method of calculating the standard deviation. The P_p_ index is determined using the following formula [26]:(12)$$P_{p}=\frac{T}{6\bar{\sigma }}$$
where $\bar{\sigma }$ is a long-term standard deviation. The P_pk_ index is determined on the basis of the following formula [26]:(13)$$P_{\mathrm{pk}}=\min\left(P_{\mathrm{pL}},P_{\mathrm{pU}}\right)$$
where:(14)$$P_{\mathrm{pL}}=\frac{\bar{X}-\mathrm{LSL}}{3\bar{\sigma }}$$
(15)$$P_{\mathrm{pU}}=\frac{\mathrm{USL}-\bar{X}}{3\bar{\sigma }}$$

The meaning of the variables in Equations (12)–(15) is the same as in Equations (8)–(11).

If the process is stable over time, the process capacity and process capability indices are almost equal.

Table 8 shows the determined actual values of the mean dispersion, $\bar{X}$, values of standard deviations σ and long-term standard deviations $\bar{\sigma }$ for individual probes and sequences. Table 9 presents the P_p_, P_pk_, C_p_, and C_pk_ indices with highlighted lowest and highest values of these indices for individual measured dimensions. Table 8 also includes highlighted corresponding values $\bar{X}$, σ, and $\bar{\sigma }$. The minimum calculated values of the P_p_, P_pk_, C_p_, and C_pk_ indices for the measurement process are still an order of magnitude greater than the values determining the measurement process using the tested probes as ideal (the last line in Table 7). This means that even in the worst case scenario, the measurement process is perfect. For the worst-case scenarios, charts were drawn to illustrate the quality of the process.

Figure 12 shows the distribution of errors of measuring the base surface Z coordinate position for sequence 4 and probe T4. Measurement errors are distributed approximately in the center of the tolerance zone at a great distance from the USL and LSL. 

Figure 13 shows the distribution of errors of measuring the ring axis TP position for sequence 4 and probe T3. In this case, the LSL is zero because the ring axis TP position error cannot be negative. In this case, the LSL is called the lower boundary (LB) [27]. Measurement errors are distributed close to the LB at a large distance from the USL.

Figure 14 shows the distribution of errors of measuring the reference ring U diameter for sequence 2 and probe T2. As before, the LSL = LB = 0 because the U error has been defined as the absolute value obtained from the difference of diameters and cannot be negative. Measurement errors are distributed close to the LB at a large distance from the USL [27].

The measurements results and graphs were prepared using Minitab (Version 14, Minitab Inc., State College, PA, USA, 2014) [28].

## 6. Discussion

The maximum obtained error in measuring the ring axis base surface Z coordinate was 2.0 µm (Figure 9) and it is an order of magnitude smaller than the 50-µm required tolerance for bearing seat face surfaces in gearbox housing (Table 6). The maximum obtained error in measuring the ring axis TP position is 5.32 µm (Figure 10). It is an order of magnitude smaller than the required 50 µm accuracy of the housing and gearbox bearing seats (Table 6). The maximum error in measuring the reference ring U diameter is 2.2 µm (Figure 11) and is an order of magnitude smaller than the required 25-µm accuracy of the bearing seat diameter (Table 6).

R&R indices (Table 5) have the highest values expressed in absolute numbers for sequence 3 (R&R values do not exceed 2.3 µm). In this sequence, all measurements were made without a tool replacement. Probably, in this sequence, there occurred an unfavorable case of mounting in the machine holder, which resulted in an increased measurement error. In sequence 4, during which the tool was mounted and removed every time, the effect of the mounting method on the results was different each time, and this contributed to an improvement in the analysis results for that sequence. The cleanliness of the base surfaces of both the probe holder and the machine spindle is therefore a key factor in the quality of measurements.

The obtained %R&R indices demonstrate that the measurement process using the RMP600 strain gauge probes in the Okuma MU6300V center should be regarded as very good, because all the percentage values of these indicators, which are related to the actual tolerance values of the individual dimensions, are significantly lower than 5%. The highest %R&R values were achieved for all tested dimensions in sequence 3, when all measurements were made without tool replacement.

The highest %AV measurement reproducibility values occurred in sequence 3 for each of the measured characteristics. They are 2.33% for Z, 3.79% for TP, and 3.51% for the U measurement (Table 5). The worst reproducibility in sequence 3 is related to the aforementioned single case of unfavorable mounting.

The highest %EV measurement repeatability values occurred in sequence 4 for each of the measured characteristics. They are 3.0% for Z, 2.92% for TP, and 4.08% for the U measurement (Table 5). The worst repeatability observed in this sequence is justified, because the probe was collected from the tool warehouse and mounted in the machine holder every time, which affected the measurement results, causing their variability.

The minimum values calculated for the measurement processes, C_p_ = 12.0; C_pk_ = 11.11; P_p_ = 10.05, and P_pk_ = 10.01 (Table 9) are an order of magnitude greater than the values determining the measurement process performed with the tested probes as ideal.

The obtained actual mean dispersion values $\bar{X}$ (Table 8) assume values of up to several micrometers, which allows to assume that there are no significant systematic measurement errors. The $\bar{X}$ values are located in the immediate vicinity of the process tolerance center in the case of symmetric tolerance (Z) and in the immediate vicinity of the LB = 0 for one-sided tolerances (TP and U). 

The maximum short-term and long-term variability of the measurement process performed with T2, T3, and T4 probes, in each analyzed case, is an order of magnitude lower than the tolerance zone for the implementation of a given geometric characteristics.

On the basis of the described test, it cannot be unequivocally stated that the length of the probe measuring element (stylus) and the shape of the measuring tip (sphere, disc) had a significant impact on the obtained values of measurement errors.

## 7. Comparative Studies

To evaluate the quality of the proposed measurement system, verification tests were conducted on 25 sample parts. They consisted of measuring the position of the axes of the holes made in these parts. The course of the study was as follows:Make holes on CNC machine in drilling and boring operations;Measurement of the position of the hole axis relative to the base hole axis on the CNC machine immediately after the hole is made; andMeasurement of the hole axis position relative to the base hole axis on a CMM.

The test parts were 50-mm thick aluminum plates, which provided them with high rigidity and eliminated the deformation resulting from the release of stresses during machining, and also eliminated the effect of the difference in clamping of the parts on the CNC machine and CMM. Figure 15 shows the errors of TP position measurements of hole axes in test parts for CNC and CMM measurements.

Visual evaluation of the data allows us to conclude that the range of measurement errors as well as their variance are similar. In order to quantitatively evaluate the obtained results, several statistical features were calculated and are presented in Table 10.

In conclusion, it can be said that the measurement errors on the CNC and CMM have similar statistical properties which confirms the validity of the implementation of measurements on the CNC machine.

## 8. Conclusions

The article contains an assessment of a measuring system consisting of a CNC machine and measuring probes. The assessment was carried out using two methods on the basis of the obtained results of the rigid body measurements, which was a reference ring with known dimensions and a known position in the machine working space. The analysis of the measurement results made it possible to obtain information about the measurement system. In turn, the reference of the measurement results to the tolerance of the target products made it possible to assess the possibility of using the measurement system for product quality control. In the light of the adopted requirements for the quality of products, it can be concluded that the measurement system is able to provide reliable measurement reports and has a potential application as an element of CDT.

Besides the virtues of reducing the human factor, the proposed approach is economically efficient. This is related to the structure of the production lines. The ratio of CNC machines to CMMs is usually from 6:1 to 10:1. Measurement stations or measurement laboratories generate a bottleneck problem in the manufacturing process [29], because parts from several CNC machine tools are waiting to be measured on a single CMM. In addition, gearbox cases must be unmounted from the CNC machine, washed at high temperature, dried, and then heat-stabilized before being delivered to the lab, considering the large size of the parts, this generates waiting times of up to several hours. The second consideration is the ratio of the price of the CMM to the cost of adding probes to CNC machines. Considering that the cost of a CMM is about 300,000 USD and the cost of a set of three probes with additional software is only 20,000 USD, the benefit of eliminating CMMs is clear. If even some of the most critical measurements need to be made on a CMM, and the majority of measurements can be made on a CNC machine, then offloading the CMM eliminates the bottleneck problem in the manufacturing process and reduces the number of CMMs.

The next stage of research will consist of the application of the measurement procedure in the entire production process of aircraft gearbox housings, both at the rough machining stage, when the manufacturing tolerances are much greater than the analyzed ones, and at the final machining stage.

## Figures and Tables

**Figure 1 sensors-21-04852-f001:**
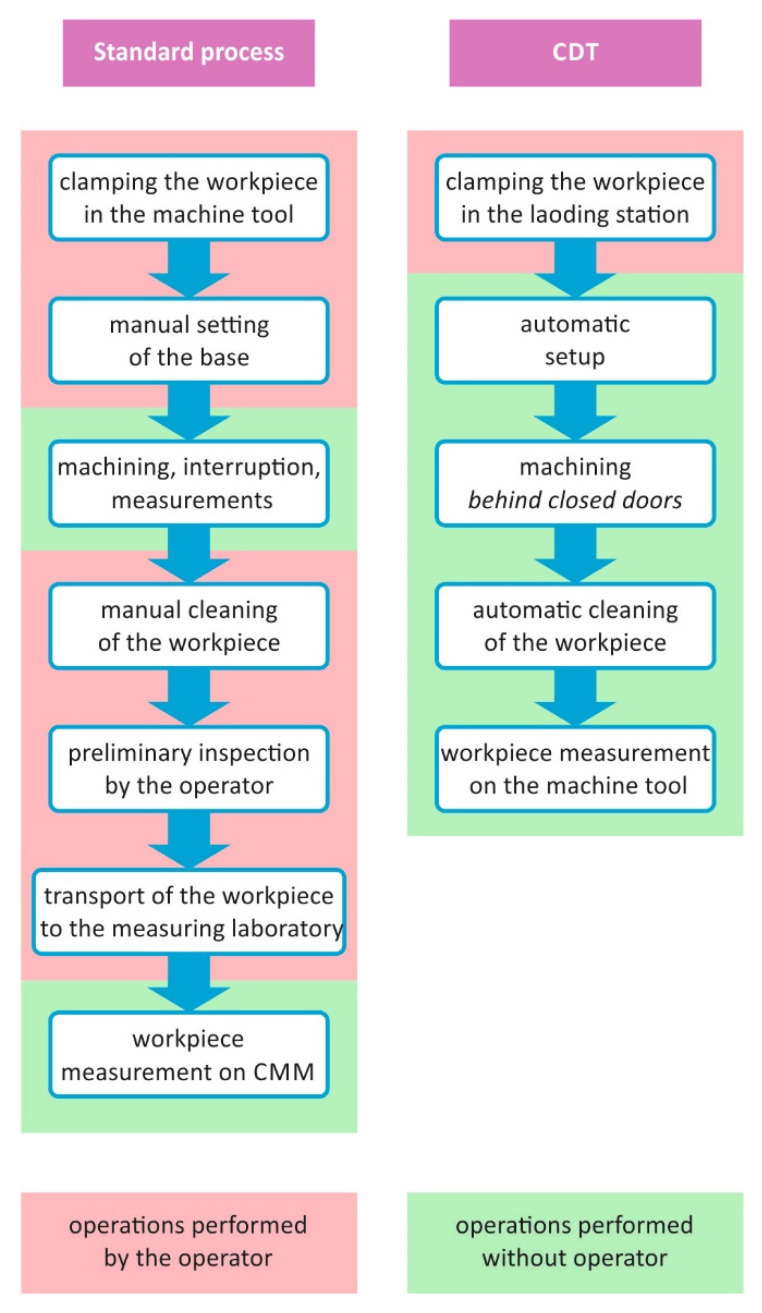
Manufacturing using the standard machining process and CDT.

**Figure 2 sensors-21-04852-f002:**
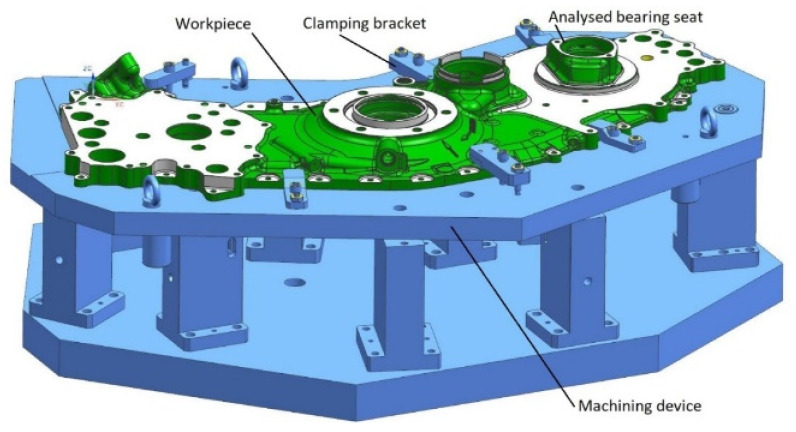
AGB housing mounted in the machining device.

**Figure 3 sensors-21-04852-f003:**
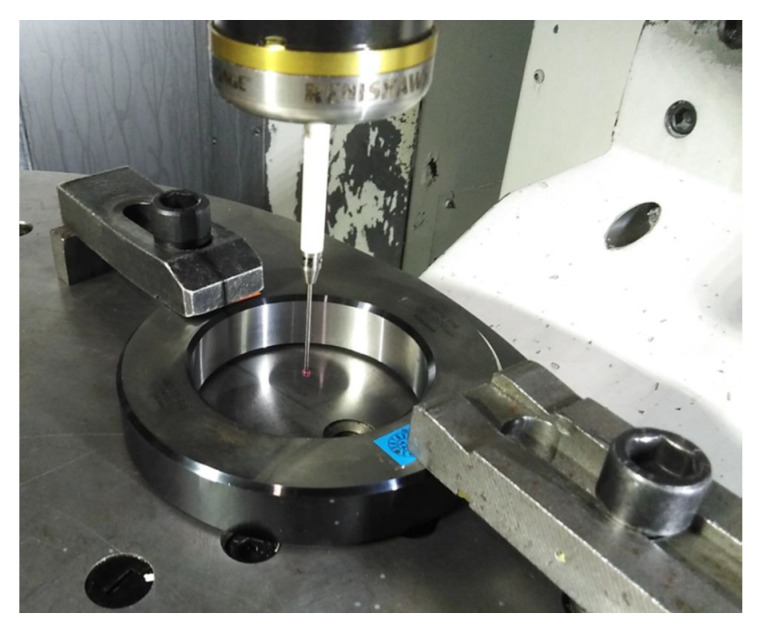
Measuring the diameter of the ring gauge on a machine tool with a standard touch probe.

**Figure 4 sensors-21-04852-f004:**
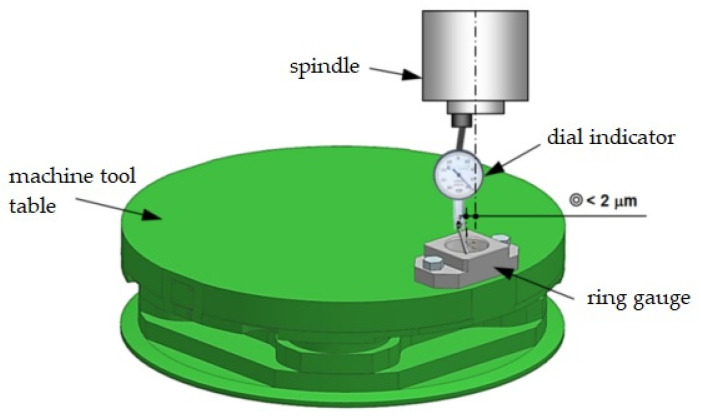
Setting the master ring gauge on the machine.

**Figure 5 sensors-21-04852-f005:**
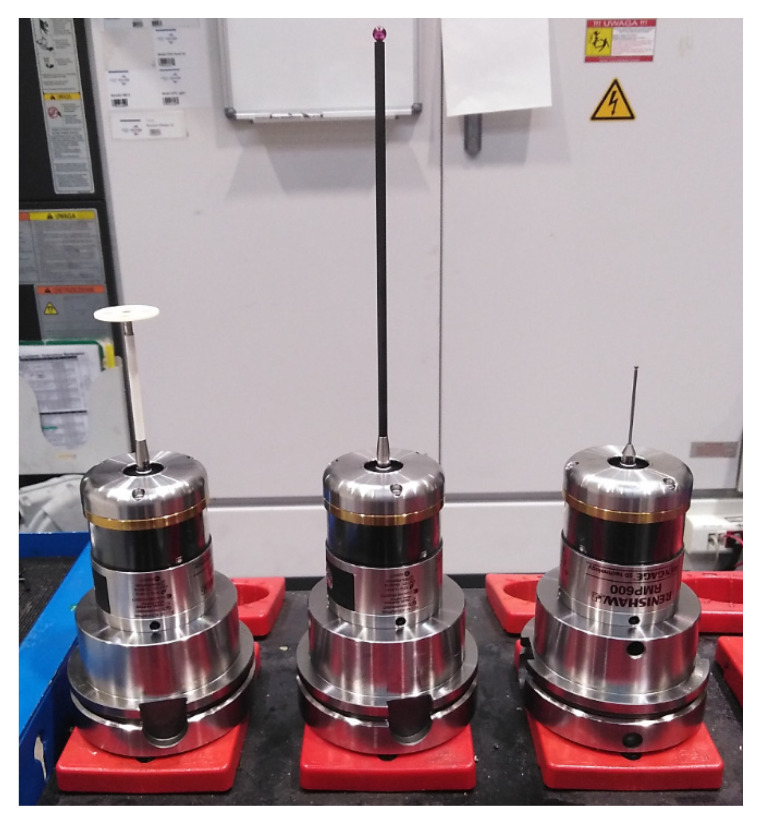
Tested T2, T3, and T4 measuring probes.

**Figure 6 sensors-21-04852-f006:**
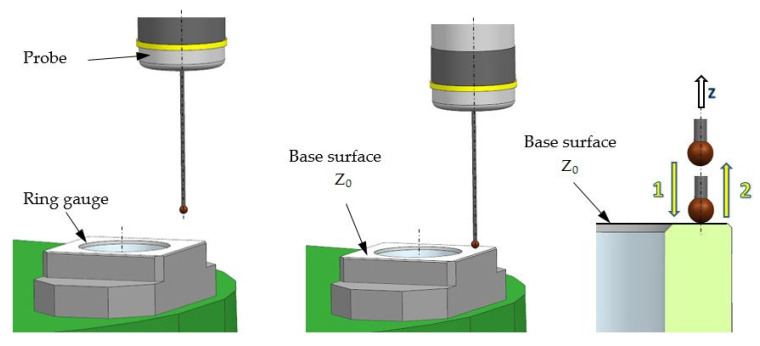
Sequence of measurement of the ring base surface coordinate Z.

**Figure 7 sensors-21-04852-f007:**
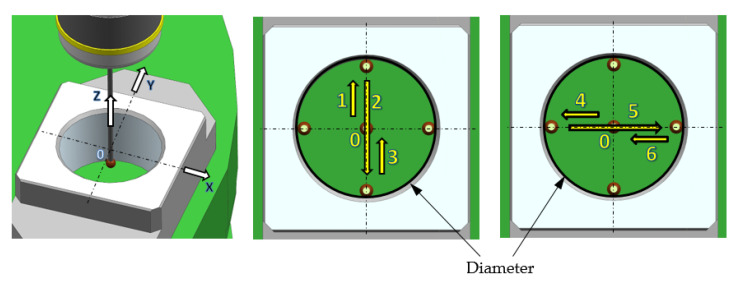
Sequence of measuring the position of the ring axis TP and the diameter ϕ value.

**Figure 8 sensors-21-04852-f008:**
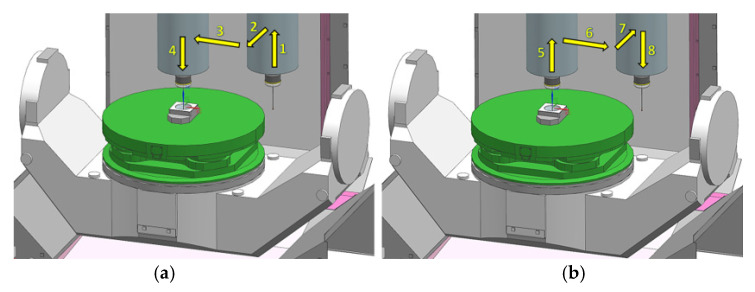
Sequence of the probe approach (**a**) Movement from the machining tool replacement point to the measurement area; (**b**) Back to the machining tool replacement point.

**Figure 9 sensors-21-04852-f009:**
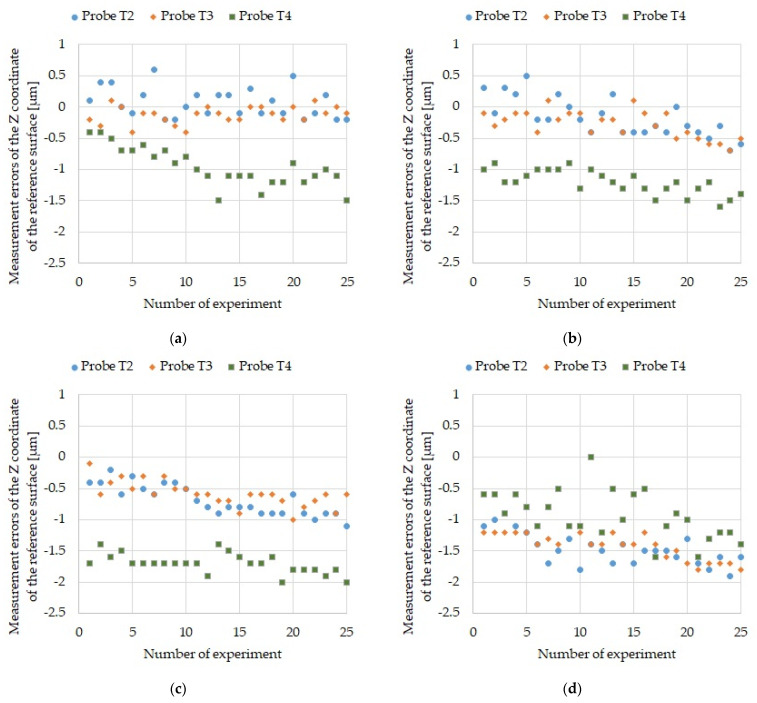
Base surface Z coordinate position measurement errors for all probes: (**a**) For sequence 1; (**b**) For sequence 2; (**c**) For sequence 3; and (**d**) For sequence 4.

**Figure 10 sensors-21-04852-f010:**
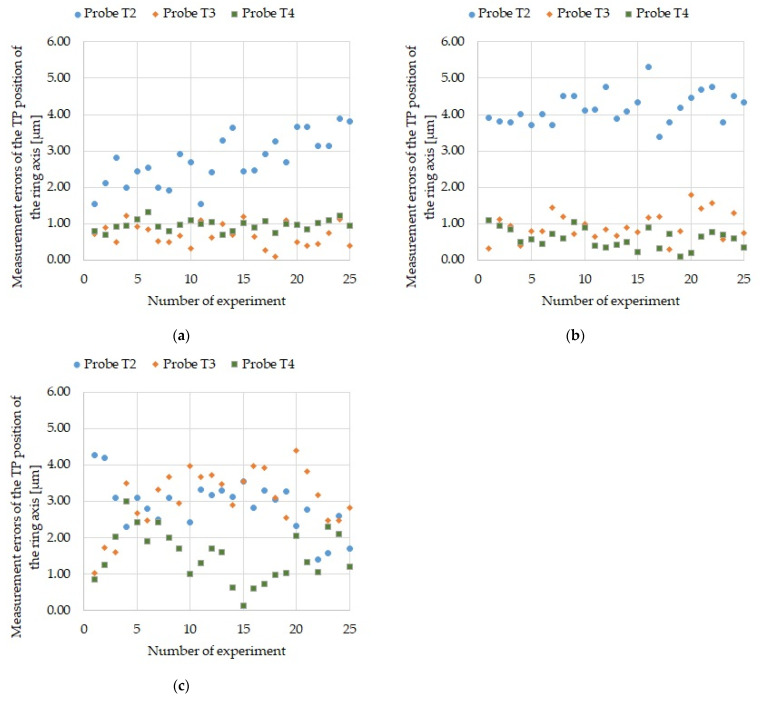
Ring axis TP position measurement errors for all probes: (**a**) For sequence 2; (**b**) For sequence 3; and (**c**) For sequence 4.

**Figure 11 sensors-21-04852-f011:**
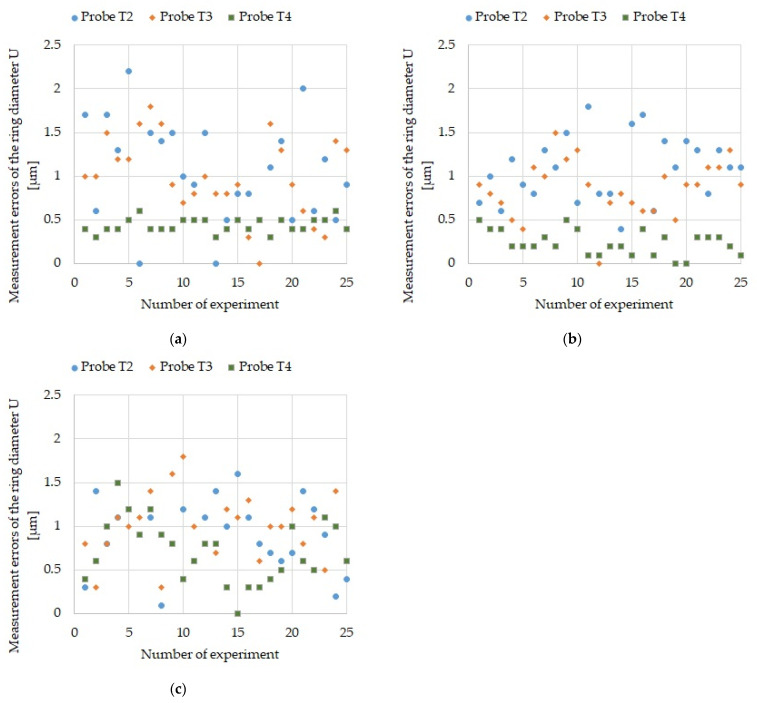
Reference ring U diameter measurement errors for all probes: (**a**) For sequence 2; (**b**) For sequence 3; and (**c**) For sequence 4.

**Figure 12 sensors-21-04852-f012:**
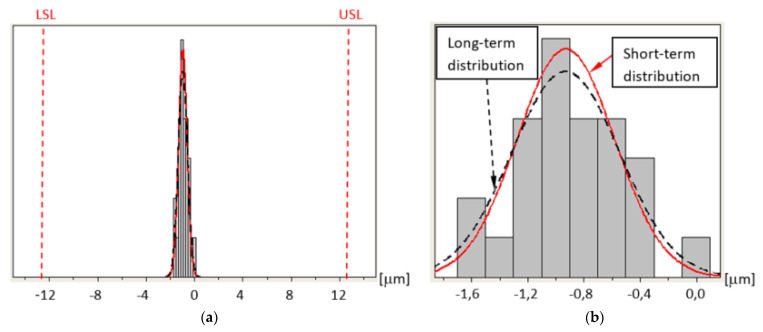
Distribution of errors of measuring the base surface Z coordinate position for sequence 4 and probe T4: (**a**) the entire tolerance zone view and (**b**) detailed view.

**Figure 13 sensors-21-04852-f013:**
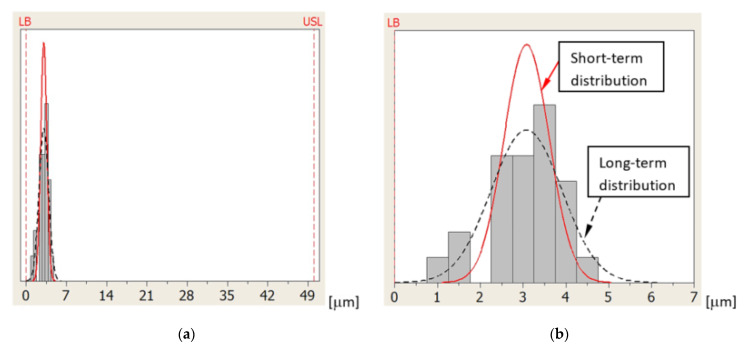
Distribution of errors of measuring the ring axis TP position for sequence 4 and probe T3: (**a**) the entire tolerance zone view and (**b**) detailed view.

**Figure 14 sensors-21-04852-f014:**
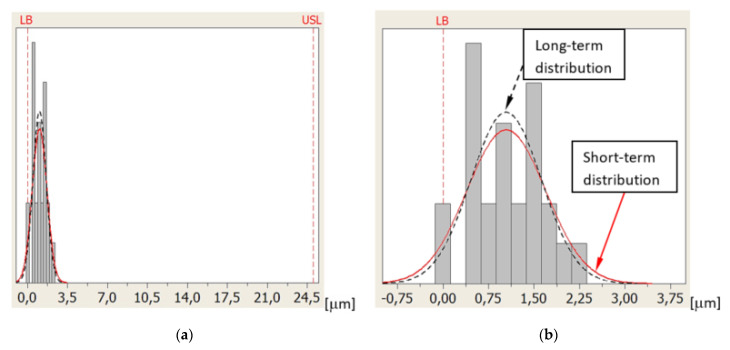
Distribution of errors of measuring the reference ring U diameter for sequence 2 and probe T2: (**a**) the entire tolerance zone view and (**b**) detailed view.

**Figure 15 sensors-21-04852-f015:**
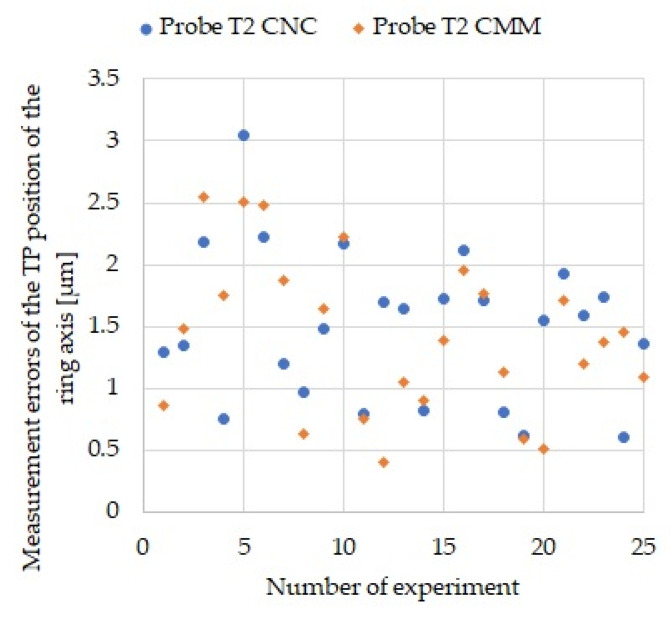
TP position errors of hole axes in test parts for CNC and CMM measurements using a T2 probe.

**Table 1 sensors-21-04852-t001:** RMP600 probe parameters.

Unidirectional Repeatability	2D Lobbing	3D Lobbing	Minimum Force Value for Probe Activity in the Following Directions
0.25 µm 2σ for stylus length up to 50 mm	±0.25 µm for stylus length up to 50 mm	±1.00 µm for stylus length up to 50 mm	X,Y = 0.2 N
0.35 µm 2σ for stylus length up to 100 mm	±0.25 µm for stylus length up to 50 mm	±1.75 µm for stylus length up to 50 mm	Z = 1.9 N

**Table 2 sensors-21-04852-t002:** Geometric characteristics of T2, T3, and T4 probe measuring tips.

Probe Number	Stylus Length (mm)	Measuring Tip Diameter (mm)	Measuring Tip Shape
T2	L = 50	2	Sphere
T3	L = 200	6	Sphere
T4	L = 75	30	Disc

**Table 3 sensors-21-04852-t003:** Maximum measurement error values for various probes and sequences, provided in µm.

Probe	Sequence 1			Sequence 2			Sequence 3			Sequence 4		
	Z	TP	U	Z	TP	U	Z	TP	U	Z	TP	U
T2	0.6	-	-	−0.7	3.89	2.2	−1.1	5.32	1.8	−1.9	4.27	1.6
T3	−0.4	-	-	−0.7	1.22	1.8	−1.0	1.80	1.5	−1.8	4.39	1.8
T4	−1.5	-	-	−1.6	1.32	0.6	−2.0	1.10	0.5	−1.6	3.00	1.5

**Table 4 sensors-21-04852-t004:** Average measurement error values for various probes and sequences, provided in µm.

Probe	Sequence 1			Sequence 2			Sequence 3			Sequence 4		
	Z	TP	U	Z	TP	U	Z	TP	U	Z	TP	U
T2	0.072	-	-	−0.168	2.76	1.04	−0.688	4.18	1.08	−1.468	2.83	0.90
T3	−0.124	-	-	−0.276	0.70	1.00	−0.588	0.94	0.86	−1.412	3.08	0.98
T4	−0.960	-	-	−1.204	0.96	0.44	−1.704	0.60	0.24	−0.928	1.50	0.71

**Table 5 sensors-21-04852-t005:** The values of R&R indices for various sequences.

	Sequence 1			Sequence 2			Sequence 3			Sequence 4		
Indicator	Z	TP	U	Z	TP	U	Z	TP	U	Z	TP	U
R_AVE_ ^1^	−0.34	-	-	−0.55	1.47	0.83	−0.99	1.91	0.73	−1.27	2.47	0.86
EV ^1^	−0.199	-	-	−0.325	0.871	0.488	−0.587	1.127	0.429	−0.750	1.459	0.510
X_DIFF_ ^1^	1.03	-	-	1.04	2.06	0.60	1.12	3.58	0.84	0.54	1.58	0.27
AV ^1^	0.548	-	-	0.547	1.080	0.306	0.582	1.894	0.439	0.245	0.789	0.103
R&R ^1^	0.583	-	-	0.636	1.388	0.576	0.827	2.204	0.613	0.789	1.659	0.521
%EV ^2^	−0.40			−0.65	1.74	1.95	−1.17	2.25	1.71	−1.50	2.92	2.04
%AV ^2^	1.10			1.09	2.16	1.22	1.16	3.79	1.75	0.49	1.58	0.41
%R&R ^2^	1.17			1.27	2.78	2.31	1.65	4.41	2.45	1.58	3.32	2.08

^1^ value provided in µm. ^2^ value provided in %.

**Table 6 sensors-21-04852-t006:** Manufacturing tolerance zones of particular elements.

Parameter	Tolerance Zone
Base surface Z coordinate position	50 µm
Ring axis TP position	25 µm
Ring U diameter error	50 µm

**Table 7 sensors-21-04852-t007:** Diagnosing the qualitative capacity of the process based on the values of the C_p_ and C_pk_ indices.

C_p_ Index Value	C_pk_ Index Value	Diagnosis of The Process Qualitative Capacity
C_p_ = 1	C_pk_ = 1	Qualitatively capable production process. Faulty product fraction F = 0.27%
C_p_ < 1	C_pk_ > 1	Qualitatively incapable production process requires refinement or tolerance expansion
C_p_ = 1.33	C_pk_ = 1.33	Production process with large qualitative capacity Faulty product fraction F = 0.006%
C_p_ = 1.66	C_pk_ < 1	Qualitatively incapable production process. The process is affected by a systematic factor. The factor must be removed.
C_p_ > 1.66	C_pk_ > 1.66	Ideal qualitative capacity of the process. Faulty product fraction F = 0.000006%.

**Table 8 sensors-21-04852-t008:** Actual mean dispersion values $\bar{X}$, standard deviation values σ, and long term standard deviations $\bar{\sigma }$, for various probes and sequences provided in µm (the minimum values of statistical features for a given measurement error are marked with green, the maximum values of statistical features for a given measurement error are marked with red).

Measurement Errors	Probe	Sequence 1			Sequence 2			Sequence 3			Sequence 4		
		σ	$\bar{\sigma }$	$\bar{X}$	σ	$\bar{\sigma }$	$\bar{X}$	σ	$\bar{\sigma }$	$\bar{X}$	σ	$\bar{\sigma }$	$\bar{X}$
Z	T2	0.255	0.241	0.072	0.217	0.313	−0.168	0.115	0.248	−0.688	0.188	0.264	−1.468
	T3	0.266	0.138	0.124	0.266	0.219	−0.276	0.177	0.207	−0.588	0.177	0.221	−1.412
	T4	0.144	0.313	−0.960	0.140	0.199	−1.204	0.107	0.158	−1.704	0.347	0.385	−0.928
TP	T2	-	-	-	0.470	0.695	2.760	0.391	0.440	4.180	0.533	0.738	2.830
	T3	-	-	-	0.339	0.313	0.702	0.379	0.387	0.943	0.531	0.829	3.078
	T4	-	-	-	0.145	0.152	0.964	0.225	0.272	0.597	0.469	0.701	1.497
U	T2	-	-	-	0.650	0.583	1.044	0.414	0.373	1.080	0.380	0.397	0.900
	T3	-	-	-	0.299	0.467	0.996	0.244	0.330	0.856	0.392	0.376	0.980
	T4	-	-	-	0.081	0.083	0.440	0.111	0.143	0.240	0.244	0.360	0.708

**Table 9 sensors-21-04852-t009:** Values of P_p_, P_pk_, C_p_, and C_pk_ indices for various probes and sequences (the minimum values of indices for a given measurement error are marked with green, the maximum values of indices for a given measurement error are marked with red).

Measurement Errors	Probe	Sequence 1				Sequence 2				Sequence 3				Sequence 4			
		C_p_	P_p_	C_pk_	P_pk_	C_p_	P_p_	C_pk_	P_pk_	C_p_	P_p_	C_pk_	P_pk_	C_p_	P_p_	C_pk_	P_pk_
Z	T2	16.35	17.26	16.25	17.16	19.12	13.28	18.86	13.10	36.39	16.80	36.39	15.87	22.12	15.80	19.52	13.95
	T3	15.67	30.26	15.52	29.96	15.67	19.04	15.33	18.62	23.51	20.14	22.40	19.19	23.51	18.87	20.85	16.74
	T4	28.92	13.32	26.70	12.30	29.68	20.95	26.83	18.93	38.90	26.31	33.59	22.72	12.00	10.81	11.11	10.01
TP	T2	-	-	-	-	17.73	11.99	33.49	22.67	21.31	18.94	39.04	34.72	15.63	11.29	29.48	21.29
	T3	-	-	-	-	24.58	26.62	48.52	52.57	21.99	21.53	43.17	42.20	15.69	10.05	29.44	18.88
	T4	-	-	-	-	57.47	54.82	112.6	107.0	37.04	30.64	73.08	60.45	17.77	11.89	34.49	23.06
U	T2	-	-	-	-	12.82	14.29	13.69	12.28	20.13	22.34	19.27	21.34	21.93	20.99	21.11	20.22
	T3	-	-	-	-	27.87	17.84	26.74	17.13	34.15	25.25	33.01	24.38	21.26	22.16	20.45	21.30
	T4	-	-	-	-	102.9	100.4	100.7	99.23	75.08	58.28	74.48	57.76	34.15	23.15	33.21	22.52

**Table 10 sensors-21-04852-t010:** Statistical features of measurement errors.

Statistical Feature	Measurement on CNC	Measurement on CMM
Mean	1.50	1.41
Standard deviation	0.59	0.62
Variance	0.36	0.40
Range	2.44	2.15

## Data Availability

Data is contained within the article.

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
