# Peer review of "Geometric Measurements on a CNC Machining Device as an Element of Closed Door Technology"

_sensors, 2021, doi:10.3390/s21144852_

Round 1

Reviewer 1 Report

The article contains a measurement process of a CNC machining tool and measuring probes for aircraft accessory gearbox housing. The measurement process requires very precise measurements and an excellent engineering job conducted during the development process. Many measurements were made, and the results were discussed well and analyzed by statistical methods, but my main question is, what is the scientific novelty of this paper?

The introduction of the paper shows the importance of the topic. However, some more citations and some comparison of the advantages and disadvantages of the novel methods can help the readers understand the difficulties and the need for novel solutions to implementing this CNC measurement system. Please, discuss the answer to this question at the end of the introduction and show some other methodologies.

The following section shows the examined geometry some information about the used materials. The used illustration is very nice. However, if it's possible to share some more information about the used materials, it can help the reader understand more the paper's difficulties.

In line 151, "each touch probe was calibrated according to Renishaw(Renishaw plc, Wot-151ton-under-Edge, UK)probe calibration procedure", please insert some citations and discuss them in 1-2 sentence the selection of the calibration procedure and discuss the advantages and disadvantages of the selected measurement tools.

In 3.1, line 168, why did you need exactly four measurement sequences? It is discussed in detail later. However, it is very hard to follow it. Please, discuss the base ideas in some sentences to help the reader to understand your measurement process before you go into the deep details. The illustrations here are nice (Fig 4-7), but the text should be structured.

Please improve the quality, at least the resolution of Figure 8-9. Is it correct to plot the error of the different measurement sequences on the same plot?

In the next parts of the paper, the evaluation of the measurement process seems correct. However, it is hard to see what is the scientific novelty of the paper, this should be highlighted, and the text should be restricted to be followed easier by the reader than it can be an interesting paper.

Reviewer 2 Report

Overview:

This submission utilized a CNC measuring system for carrying out automatic geometric measurements in the context of closed-door machining. To evaluate the quality of the obtained measurements, the authors utilized a variety of statistical methods.

In my opinion, the presented methodology and the paper’s subject are seemingly interesting. Since the paper’s subject falls within the scope of Sensors, I encourage the authors to improve the quality of the paper according to the comments below.

Comments:

  • The paper is well-written, although I advise the authors to proofread the paper again since it contains some errors. For example:
    • Please revise the caption of Table 3, since it is not written in English.
    • In line 109, please replace “Manufacture” with “Manufacturing” or “Production”.
  • In the paper, the authors failed to review in length the vast research area of measuring systems. In this regard, it is advisable to include a literature review section that acknowledges the previous works relevant to the paper’s topic and clearly illustrates the motivation of the present paper.
  • Seemingly, the authors utilized an on-machine probing system for measuring the geometric features of aircraft accessory gearbox housing. Considering that, I am not intuitively convinced of the paper’s novelty, since the on-machine measuring systems have been widely utilized and evaluated in the literature. Hence, the authors should elaborate on the paper’s novelty and contribution in the addressed research area.
  • In the experimental analysis section, the authors assessed the measurements obtained by the proposed system using statistical techniques, such as the statistical process control (SPC) method. This analysis is not sufficient to validate the efficiency of the presented measuring system in the context of machining processes. Therefore, it would be useful to evaluate the performance of the implemented measurement system against the widely-utilized measurement tools, such as coordinate measuring machines (CMM).
  • It is not clear whether the proposed approach can be considered as a cost-effective solution for CNC machining companies. It would be substantially helpful to elaborate on that matter by providing a real-world example.

Recommendation:

Overall, I believe that the aforementioned paper can be considered for publication in Sensors, but not in its current form. Based on the comments above, my recommendation for this submission is MINOR REVISION.

Round 2

Reviewer 1 Report

The authors answered all of my questions.